# OpenReview forum: "Hierarchical Self-Supervised Graph Contrastive Learning: Capturing Multi-Scale Structural Information"
_ICLR.cc/2025/Conference — Submitted to ICLR 2025_

### Official Review · Reviewer_xmDx · 2024-10-24

**Soundness:** 2
**Presentation:** 2
**Contribution:** 2
**Rating:** 3
**Confidence:** 4

**Summary:**

This paper introduces a hierarchical self-supervised graph contrastive learning method, that uses graph augmentation strategies to generate multi-scale views for self-supervised learning.

**Strengths:**

1. The use of multi-granularity contrastive learning in the field of graph self-supervised learning is common and has been proven effective.
2. The paper provides a clear introduction to common graph augmentation methods.

**Weaknesses:**

1. The paper lacks significant innovation. Utilizing contrastive learning at the node, subgraph, and graph levels to achieve self-supervised learning on graphs is a very common and well-established approach. The authors have not provided further theoretical or experimental insights beyond existing methods.
2. The authors do not clearly articulate the motivation for their method. Given the extensive research and numerous existing approaches in graph self-supervised learning, it is unclear why this particular method is necessary or advantageous.
3. The datasets used in the experiments are small in scale and limited in number. Additionally, the comparative methods are outdated, which fails to demonstrate the effectiveness and advancement of the proposed approach.

**Questions:**

In addition to the Weakness, I have the following concerns.
1. Given that contrastive learning at the node, subgraph, and graph levels is common and well-researched, what is the research motivation of this paper? What issues remain unresolved by previous studies?
2. The self-supervised methods compared in this paper are from three years ago. It should include comparisons with more recent graph self-supervised methods, including but not limited to: GraphMAE2[1], S2GAE[2], GiGaMAE[3], AUG-MAE[4], etc.\
[1] Graphmae2: A decoding-enhanced masked self-supervised graph learner. WWW2023. \
[2] S2GAE: self-supervised graph autoencoders are generalizable learners with graph masking. WSDM2023.\
[3] Gigamae: Generalizable graph masked autoencoder via collaborative latent space reconstruction. CIKM2023.\
[4] Rethinking Graph Masked Autoencoders through Alignment and Uniformity. AAAI2024.

---

### Official Review · Reviewer_9t1t · 2024-10-30

**Soundness:** 2
**Presentation:** 2
**Contribution:** 1
**Rating:** 3
**Confidence:** 4

**Summary:**

This paper introduces a hierarchical self-supervised learning framework for graph neural networks that aims to learn better node representations by capturing structural information at multiple scales. The framework performs contrastive learning simultaneously at three levels: node-level (comparing representations of the same node across different augmented views), subgraph-level (contrasting embeddings of subgraphs with the same central node), and graph-level (maximizing agreement between node embeddings and global graph representations). The method uses adaptive graph augmentation strategies, including node feature masking, edge perturbation, and subgraph sampling, to generate different views while preserving essential graph properties. Through extensive experiments on benchmark datasets including Cora, Citeseer, PubMed, and Reddit, the authors demonstrate that their approach outperforms both supervised and self-supervised baseline models in node classification tasks, particularly in scenarios with limited labeled data.

**Strengths:**

1. The paper presents a systematic integration of multi-scale contrastive learning techniques for graphs. Sections 3.7-3.10 offers a coherent organization of node, subgraph, and graph-level contrasts.

2.  The paper clear presentation, supported by well-designed visual aids throughout. Section 3 provides a comprehensive framework overview, anchored by Figure 1 which effectively illustrates the full pipeline from graph augmentation through hierarchical contrastive learning.

3. The experimental evaluation section demonstrates consistent but modest improvements over existing baselines across standard benchmark datasets. The authors have done a commendable job in organizing the experimental analysis. Section 5's results are presented through a series of well-structured tables and ablation studies that systematically validate each component's contribution.

**Weaknesses:**

1.The paper's primary limitation lies in its theoretical foundation. While Section 3 presents a hierarchical framework combining node, subgraph, and graph-level contrasts, it lacks rigorous analysis of why these three levels work well together.

2. A significant weakness lies in the paper's positioning relative to existing hierarchical approaches in graph representation learning [1,2]. While Section 2 reviews general graph learning methods, it fails to adequately discuss existing hierarchical frameworks.

3. The framework's design appears to be a straightforward combination of existing techniques rather than a novel architectural innovation. The three-level contrast described in Section 3.7-3.10 essentially stacks parallel objectives without exploring meaningful interactions.

4. While the paper demonstrates results on traditional benchmark datasets (Cora, Citeseer, PubMed, and Reddit), it misses crucial evaluations on modern, more challenging benchmarks like the Amazon datasets.

[1] Zhang, Haonan, et al. "Multi-Scale Self-Supervised Graph Contrastive Learning With Injective Node Augmentation." IEEE Transactions on Knowledge and Data Engineering 36.1 (2023): 261-274.

[2] Yan, Hao, et al. "Hierarchical graph contrastive learning." Joint European Conference on Machine Learning and Knowledge Discovery in Databases. Cham: Springer Nature Switzerland, 2023.

**Questions:**

1. What is the theoretical justification for combining node, subgraph, and graph-level contrasts? Is there any analysis showing why these three levels work better together than individually?

2. For the subgraph-level contrast, how do you handle the varying sizes of subgraphs? Are there any specific pooling strategies that work better than others?

3. How sensitive is the performance to the choice of loss weights (α, β, γ) in the composite loss function? Could you provide more detailed ablation studies on these parameters?

---

### Official Review · Reviewer_3pLC · 2024-11-02

**Soundness:** 2
**Presentation:** 2
**Contribution:** 1
**Rating:** 1
**Confidence:** 4

**Summary:**

This paper introduces a self-supervised graph learning framework that enhances node representations without labeled data. By using a hierarchical contrastive approach at node, subgraph, and graph levels, it captures detailed structural information. Tested on datasets like Cora and PubMed, the method outperforms traditional models in low-label scenarios, proving effective for applications where labeled data is scarce, like social and biological networks.

**Strengths:**

1. The proposed methods seem to be technically sound.
2. Experimental results show that the proposed method could outperform certain baselines.

**Weaknesses:**

1. The novelty of the proposed method is limited.
2. The comparison baselines are outdated, the most recent one was published in 2020.
3. The overall presentation should be further improved.

**Questions:**

N/A

---

### Official Review · Reviewer_fWP4 · 2024-11-03

**Soundness:** 1
**Presentation:** 2
**Contribution:** 1
**Rating:** 3
**Confidence:** 5

**Summary:**

This paper introduces a hierarchical self-supervised graph contrastive learning framework to enhance node representations in GNNs using unlabeled data. By applying contrastive learning at the node, subgraph, and graph levels, the method aims to capture both local and global structural information. Experiments on benchmark datasets (Cora, Citeseer, PubMed, Reddit) reportedly show improved performance over baseline methods.

**Strengths:**

1. Hierarchical Contrastive Framework: The concept of contrastive learning at multiple levels (node, subgraph, graph) is an interesting approach to capture structural information comprehensively.
2. Easy to Follow: The paper presents the framework and experimental setup in a clear, structured manner, making it accessible and straightforward to understand.

**Weaknesses:**

1. Lack of Methodological Coherence: The connection between motivation and the proposed method is weak. The rationale behind the hierarchical approach and augmentation choices lacks clarity, making the design appear arbitrary rather than well-motivated.
2. Limited Novelty: The paper does not offer a clear innovative aspect in terms of adaptive strategies or augmentations. The “adaptive” nature claimed in the augmentation strategy is not substantiated, as the approach merely applies known graph augmentations without introducing novel adaptive mechanisms.
3. Overstated Claims: Statements such as "opening new avenues" are unsupported by either methodological novelty or empirical evidence. The work's impact appears overstated relative to its actual contributions.
4. Outdated Baselines: The paper only compares its results against outdated baseline models, limiting the strength of the findings, as more recent models may have advanced the field in ways not considered by this work.

**Questions:**

How does the proposed "adaptive" graph augmentation strategy differ from conventional augmentation techniques? Could the authors elaborate on any adaptive mechanism involved?

---

### Official Review · Reviewer_aYVT · 2024-11-08

**Soundness:** 2
**Presentation:** 3
**Contribution:** 1
**Rating:** 3
**Confidence:** 5

**Summary:**

The paper proposes a methodology for hierarchical graph contrastive learning. The main idea is to perform data augmentation at various resolution levels – including node, edge, subgraph, and global levels. Then, different contrastive learning functions are used at each of those levels, which are combined in a composite loss function. The performance of the model has been evaluated in standard benchmark graphs compared against four baseline models.

**Strengths:**

- The idea of leveraging multiple resolution levels within a contrastive learning framework is interesting.

**Weaknesses:**

- To begin with, the related work needs better organization. Currently, in Sec. 2.2 with the title ‘Deep Graph Infomax, ’ various other methods are discussed. A different organization, possibly by grouping methods based on the principles they follow, would be helpful.
- Table 1 compares the proposed approach with baseline methods regarding the structural properties that are taken into account. While this is interesting, several strong baselines are excluded, while the most recent approach (MVGRL) was published several years ago (2020). The table needs to be enhanced with more recent approaches to the problem.
- Looking at the related literature, several other works aim to perform hierarchical contrastive learning on graphs. However, none of these methodologies are being discussed in the paper. I would suggest the authors discuss this point and clearly indicate how different the proposed approach is.
- While the idea is interesting, the novelty of the methodology is limited to a combination of existing loss functions at different granularity levels.
- Another major weakness of the paper has to do with the baselines used in the study. In fact, only four methods have been used with the most recent one being MVGRL (2020) and GRACE (2020). This part of the paper needs significant enhancement.

**Questions:**

Please see the Weaknesses section.

---

### Meta-Review · Area_Chair_wPWj · 2024-12-17

**Metareview:**

The paper proposes a hierarchical contrastive learning approach for graph representation, combining node, subgraph, and graph-level contrasts. However, reviewers identified several key weaknesses. The novelty of the methodology is limited, with the proposed approach mainly combining existing loss functions and techniques at different granularities without introducing meaningful innovations. The rationale behind the hierarchical design and the adaptive graph augmentation strategy is underexplained, making the method's motivation unclear. Additionally, the baseline comparisons are outdated, focusing on models from 2020. The paper also fails to position itself clearly within the context of recent hierarchical approaches. Given these limitations, the overall contribution appears incremental rather than groundbreaking, leading to the decision to reject the paper.

**Additional Comments On Reviewer Discussion:**

The authors did not reply.

---

### Decision · Program_Chairs · 2025-01-22

Reject